# RETRACTED: Structure–Activity Relationship of Synthetic Linear KTS-Peptides Containing Meta-Aminobenzoic Acid as Antagonists of α1β1 Integrin with Anti-Angiogenic and Melanoma Anti-Tumor Activities

**DOI:** 10.3390/ph17050549

**Published:** 2024-04-24

**Authors:** Majdi Saleem Naamneh, Tatjana Momic, Michal Klazas, Julius Grosche, Johannes A. Eble, Cezary Marcinkiewicz, Netaly Khazanov, Hanoch Senderowitz, Amnon Hoffman, Chaim Gilon, Jehoshua Katzhendler, Philip Lazarovici

**Affiliations:** 1School of Pharmacy Institute for Drug Research, Faculty of Medicine, The Hebrew University of Jerusalem, Jerusalem 9112002, Israel; majdi.naamneh@mail.huji.ac.il (M.S.N.); michal.klazas@mail.huji.ac.il (M.K.); amnonh@ekmd.huji.ac.il (A.H.); katzhe@cc.huji.ac.il (J.K.); 2VINČA Institute of Nuclear Sciences, National Institute of the Republic of Serbia, University of Belgrade, Mike Petrovića Alasa 12–14, 11000 Belgrade, Serbia; momict@gmail.com; 3Institute of Physiological Chemistry and Pathobiochemistry, University of Münster, Waldeyer-Str. 15, 48149 Münster, Germany; juliusg@gmail.com (J.G.); johannes.eble@uni-muenster.de (J.A.E.); 4Debina Diagnostics Inc., 33 Bishop Hollow Rd., Newtown Square, PA 19073-3211, USA; cmarcink@temple.edu; 5Department of Chemistry, Bar Ilan University, Ramat-Gan 5290002, Israel; netalyk@gmail.com (N.K.); hsenderowitz@gmail.com (H.S.); 6Institute of Chemistry, The Hebrew University of Jerusalem, Jerusalem 9190401, Israel; chaimgilon@gmail.com

**Keywords:** MABA, KTS, peptide synthesis, modeling, conformation, stability, integrin α1, cell adhesion, angiogenesis, melanoma tumor

## Abstract

To develop peptide drugs targeting integrin receptors, synthetic peptide ligands endowed with well-defined selective binding motifs are necessary. The snake venom KTS-containing disintegrins, which selectively block collagen α1β1 integrin, were used as lead compounds for the synthesis and structure–activity relationship of a series of linear peptides containing the KTS-pharmacophore and alternating natural amino acids and 3-aminobenzoic acid (MABA). To ensure a better stiffness and metabolic stability, one, two and three MABA residues, were introduced around the KTS pharmacophore motif. Molecular dynamics simulations determined that the solution conformation of MABA peptide **4** is more compact, underwent larger conformational changes until convergence, and spent most of the time in a single cluster. The peptides’ binding affinity has been characterized by an enzyme linked immunosorbent assay in which the most potent peptide **4** inhibited with IC_50_ of 324 ± 8 µM and 550 ± 45 µM the binding of GST-α1-A domain to collagen IV fragment CB3, and the cell adhesion to collagen IV using α1-overexpressor cells, respectively. Docking studies and MM-GBSA calculations confirmed that peptide **4** binds a smaller region of the integrin near the collagen-binding site and penetrated deeper into the binding site near Trp1. Peptide **4** inhibited tube formation by endothelial cell migration in the Matrigel angiogenesis in vitro assay. Peptide 4 was acutely tolerated by mice, showed stability in human serum, decreased tumor volume and angiogenesis, and significantly increased the survival of mice injected with B16 melanoma cells. These findings propose that MABA-peptide **4** can further serve as an α1β1-integrin antagonist lead compound for further drug optimization in angiogenesis and cancer therapy.

## 1. Introduction

Integrins represent the major multi-functional cell-adhesion transmembrane receptors that serve as extra-cellular matrix–cytoskeletal linkers and transducers in biochemical and mechanical signals between cells and their environment in different physiological and pathological conditions. Every integrin is composed of an α-subunit and a β-subunit organized in a noncovalent complex, and 18 α- and 8 β-subunits create 24 practically well-defined heterodimeric transmembrane receptors. Their functions are regulated by a delicate balance between active and inactive conformations, which are converted between each other via multiple mechanisms, including protein–protein phosphorylation and interactions with adapter proteins (e.g., kindlins), and trafficking [1]. Due to their exposure on the cell surface, integrins have been intensively investigated as pharmacological targets [2]. However, given the difficulties in generating small molecule full antagonists, integrin-targeting therapeutics have been a challenge [3], with ongoing research efforts towards drug development for different unmet clinical needs [4,5]. Upon ligand binding, integrins cluster and enroll by their cytoplasmic domains cytoskeletal, adaptor, and signaling proteins, thus in the end forming focal adhesions that anchor the cell to the immobilized extracellular matrix ligands, allowing bidirectional signals into the cell [6]. From the focal adhesion sites, signal pathways diverge and regulate adhesion, migration, proliferation, and survival [7]. Integrins play an important role in the angiogenic process, the generation of new capillaries, and in physiological as well as in pathological blood vessel formation [8]. The vascularization of tumors implanted into integrin α1-knock out mice was decreased, with a reduction in capillary size and number [9]. Both integrins, α1β1 and α2β1, are collagen receptors and contribute to the metastatic process, as evidenced by the observation that inhibitors or genetic ablation of these integrins [10], such as in integrin α1-knock out mice [11], markedly reduced tumor volume in comparison with wild-type mice. These preclinical findings indicate the important role of α1β1 in providing critical support for endothelial cell migration and tumor angiogenesis [12,13], prompting peptidomimetic antagonist development towards this collagen binding integrin [14], which could be instrumental as an anti-angiogenic and anti-cancer drug [4].

Snake venom disintegrins are the most potent natural and selective inhibitors of integrins. KTS-disintegrins, such as obtustatin [15,16] and viperistatin [17,18], uniquely contain the Lys-Thr-Ser tripeptide motif and bind selectively to the α1β1 integrin [19,20]. This KTS sequence motif (Lys-Thr-Ser tripeptide selective motif for binding collagen-integrin receptors) has been used as a pharmacophore for developing different linear and cyclic peptide antagonists of α1β1 integrins with anti-angiogenic and anti-tumor activity [14,21]. In an attempt to overcome the proteolytic susceptibility of KTS-peptides, we sought to investigate in the present study the structure–function activities of KTS-linear peptides incorporating 3-aminobenzoic acid (MABA).

3-Aminobenzoic acid (also known as meta-aminobenzoic acid or MABA) consists of a benzene ring substituted with an amino group and a carboxylic acid with the molecular formula H_2_NC_6_H_4_CO_2_H. Because synthetic amino acids, such as MABA, are not found in natural proteins, most enzymes do not recognize them for proteolytic breakdown. In synthetic peptides, however, these elementary units can be included into peptides for a variety of pharmaceutical applications, including conformational constraints, pharmacologically active compounds, assembly of diverse combinatorial libraries, and in powerful molecular scaffolds. The introduction of MABA into acyclic tripeptides [22] leads to reduced conformational flexibility and inhibits the formation of intra-molecular hydrogen bonds that could affect binding pockets by folding-induced selectivity [23,24]. The aromatic subunits also allow the insertion of further substituents and can mediate interactions with corresponding binding receptor partners [25]. Considering the oral delivery route in MABA peptide-drug development, it is important to notice that MABA is transported in the gastro-intestinal tract by a carrier-mediated transport system, important for its absorption and metabolic characteristics [25]. Active ion transporters have been developed from acyclic octapeptides comprising MABA [26] and MABA monopeptide esters and showed promising antimicrobial activity towards gram-positive bacteria [27]. Different MABA analogs showed anti-coagulant and anti-platelet activities [28], hexapeptides with MABA were characterized as potential isosteres of the Val-Thr dipeptide unit, and a peptidomimetic incorporating MABA proved to effectively inhibit oligo-saccharyl transferase, responsible for asparagine-linked glycosylation in the lumen of the endoplasmic reticulum [29].

With this background, KTS-linear peptides targeting α1β1-integrin, with alternating natural and unnatural amino acid sequences, the latter of which were MABA, were designed, synthesized, modeled, and characterized in vitro and in vivo, hypothesizing that MABA will make the peptide metabolically stable, with reduced conformational flexibility, while preserving its integrin binding properties and biological activities.

## 2. Results

### 2.1. Design and Synthesis of MABA-Peptides

To investigate the effect of number and position of MABA units, eight peptide analogs containing the KTS binding motif to α1β1 integrin were constructed according to standard Fmoc (fluorenyl-methoxy-carbonyl) SPPS (solid phase peptide synthesis) conditions, employing Wangresin and HBTU/HOBt (2-(1H-benzotriazole-1yl)-1,1,3,3-tetramethyluronium hexafluorophosphate/1-hydroxybenzo-triazole)as coupling reagents (Appendix A). All the MABA-peptide sequences were derived with some modifications from the disintegrin viperistatin sequence at positions 20–36 (^20^W-*KTS*-R-TSHYCTGKSCDG^36^) [1,4,17]. In all cases, the KTS-pharmacophore responsible for the selective binding to the α1β1 integrin was preserved, while cysteine at position 34 was replaced with serine, and the cysteine at position 29 was omitted (peptide 8), or replaced by MABA unit, or glycine. Peptide **1**, indeed, includes these two changes. One to three MABA units at different positions around the KTS motif were introduced in order to form a bent in the peptide chain and to restrict conformational diversity. In peptide **2**, the MABA element was inserted between the KTS and the RTS motifs, while the amino acids threonine and glycine at positions 30 and 31 were omitted. The sequence of peptide **3** is similar to that of peptide **2** with one distinction: an additional KTS unit was attached to the amino terminal end of the sequence. Peptide **4** includes two MABA units. One is located as in peptide **2** and the second was inserted between positions 24 and 25 of viperistatin-derived peptide **8**. The amino acid sequence is the same as in peptide **2**. Peptide **5** also includes two MABA elements at the same positions as in peptides **1** and **2**; however, the amino acid sequence corresponds to the sequence of peptide **1**. Peptide **6** is like peptide **4** and incorporates two changes, one the exchange of the serine at position 23 for arginine, and the second the amino acid sequence, which is similar to that of peptide **1**. Peptide **7** is composed of three MABA units. Two are in the same positions as in peptide **6**, and the third is as in peptide **1**. Peptide **8** is the original sequence motif of viperistatin lacking the cysteine at position 29 (Appendix A). All peptides were synthesized by SPPS applying Fmoc chemistry and purified by preparative HPLC (high-pressure liquid chromatography) to >95% homogeneity (Appendix A) and their structure confirmed by analytical LC/MS (liquid chromatography/mass spectrometry) (Appendix A).

### 2.2. MABA-Peptides Inhibit Integrin Binding and Cell Adhesion

In a previous study, linear obtustatin and viperistatin peptidomimetics containing the KTS motif were synthesized and their anti-angiogenic activity was investigated [14], representing the starting point for the synthesis of small peptide lead compounds toward the collagen-binding integrins (α1β1, α2β1). As shown earlier, Thr^22^ appeared to be the most crucial residue for integrin binding, whereas substitution of the flanking lysine or serine residues for alanine caused a less pronounced decrease in the anti-α1β1 integrin activity of the disintegrin [20]. Moreover, the presence of Arg^24^ next to the K^21^T^22^S^23^ motif strongly increases the binding to the α1β1integrin, suggesting that the positively charged side chain of Arg forms a better contact than the alkyl chain of Leu within the ligand-binding pocket of the α1β1integrin [20]. To investigate the influence of MABA on the α1β1integrin receptor binding and cell adhesion, a library of eight peptides has been examined by competition for CB3 (a collagen IV fragment) binding to the GST-α1-A domain using ELISA (enzyme-linked immunosorbent assay), and by cell adhesion to immobilized collagen IV using α1-overexpressor cells. Peptides **1–8** were incubated with the GST-linked α1-A domain and allowed to bind to the immobilized CB3 (collagen IV fragment). The portion of the bound recombinant α1-A domain accounted for the inhibitory potential and the recognition ability of these peptides (Table 1). Peptides **1–8** dose dependently inhibited binding of GST-α1-A domain, with modest IC_50_ (Inhibitory concentration 50%) values, in the 100s micromolar range (Table 1). Peptides **3** and **4** were the most potent, albeit with low avidity of IC_50_ 442 ± 12 and 324 ± 8 µM, respectively. These findings suggest that peptides **1–8** recognize with low potency a sequence motif in the α1-A domain. The ability of peptides **1–8** to inhibit the adhesion of α1-K562 (human erythroleukemic cells, overexpressing α1β1 integrin) to immobilized collagen IV was also evaluated. This cellular model was widely used to characterize potential antagonists of α1β1 integrin [14,17,30]. In these experiments, the cells were applied on plates coated with different extracellular matrix proteins and allowed to adhere before detection of the adhered cells number. Under these conditions, no significant cell adhesion to BSA (bovine serum albumin)-coated plates (negative control) or to non-specific substrate-coated plates was observed. The results are summarized in Table 1. Peptide **3** retained a notable inhibition of the adhesion of the cells to immobilized collagen IV, with an IC_50_ of >900 μM. All the other compounds poorly affected the adhesion of the cells to immobilized collagen IV (Table 1). Peptide **4** exhibited the highest potency as an inhibitor of the cell adhesion mediated by α1β1 integrin, with an IC_50_ of 550 ± 45 µM, similar to the value obtained by the competitive ELISA binding assay. The selectivity of the inhibitory effects of MABA-peptide **4**, compared to Viperistatin, on various integrins was confirmed in cell adhesion assays using cell lines expressing different major integrin subunits (Appendix A). Noteworthy, peptide **4**, like viperistatin, showed an outstanding selectivity towards the α1β1 integrin, about five-fold lower inhibitory activity toward α2β1, and did not show any activity toward other integrins, such as α6β1, α5β1, α4β1, α9β1, αvb3 and αIIbb3. These findings demonstrate that target integrin selectivity can be preserved by switching from proteins to small MABA- peptidomimetics, but further optimization is required to improve their potency.

### 2.3. In Vitro Metabolic Stability of MABA-Peptides

To investigate the proteolytic susceptibility of the synthetic peptides, the degradation of intact peptides **2** (one MABA unit), **4** (two MABA units) and **8** (linear peptide) (Table 1) incubated in rat serum at 37 °C for different periods of time, was followed by LC/MS. The results are presented in Figure 1. Peptide **8**, lacking MABA was rapidly degraded, with a half-life of about 20 min. In contrast, peptides **4** and **2**, containing two and one MABA respectively, were stable up to 60 min. Therefore, the water solubility and metabolic stability enable the MABA-peptides to circumvent some challenges to the usage of peptides as candidates in drug development [31].

### 2.4. In Vitro Anti-Angiogenic Effect of the MABA-Peptides

Angiogenesis is a cellular process characterized by endothelial cell migration, invasion, and assembly into capillaries. In vitro endothelial tube formation assays are performed using primary human umbilical vein endothelial cells that, upon applying to the Matrigel matrix, reorganize to create a tube-like structure, which may be employed as a model for studying some aspects of in vitro angiogenesis [32]. Number of nodes, junctions, meshes and total tube length were the parameters chosen for quantification of the anti-angiogenic effects of the MABA-peptides. Nodes function assesses points where branches of the network converge, a junction is an assembly of nodes, a mesh is an area entirely closed by neighboring branches, and tubes’ length is the total area of skeletonized network branches [32,33]. HUVEC (human umbilical vein endothelial cells) plated on Matrigel supplemented with the VEGF (vascular endothelial growth factor) (30 ng/mL) formed a capillary-like network within 16 h (Figure 2). In the presence of peptides **3** and **4** (500 μM), the extent of tube formation evaluated by all parameters was significantly reduced in comparison to that in the cells treated with the vehicle alone (containing up to 1% DMSO dissolved in cell culture medium). Interestingly, peptide **4** exhibited the greatest inhibitory effect in blocking VEGF-induced angiogenesis. The number of junctions, nodes, meshes and tube length were reduced by 70–80% (*p* ≤ 0.01 and 0.001 vs. control). The minimal concentration of peptides **3** and **4** yielding a complete inhibition of endothelial morphogenesis on Matrigel was about 200 μM, whereas the other peptides **1**, **2**, **6**, **7** and **8** in millimolar concentrations did not cause any significant reduction in the level of angiogenesis (Appendix A).

### 2.5. Anti-Tumor Effect of the MABA-Peptides in a Murine Melanoma Model

To evaluate their anti-tumor efficacy, the MABA-peptides were administered systemically by intra-peritoneal injection, three times a week, for three weeks with a cumulative dose of 90 mg/kg in C57BL/6 mice bearing B16-F10 (a melanoma tumor cell line) tumors. Figure 3A shows the mean tumor volume of mice treated with MABA peptide-4 as compared to saline-treated mice. A significant (*p* < 0.01 vs. untreated sick mice) delay of tumor growth was observed in the peptide **4** treated mice at 14 and 16 days after tumor cells’ injection. Body weights were monitored twice a week throughout the study and were not statistically different between the groups at any time (25–30 g), indicating lack of toxicity. Figure 3B shows the Kaplan–Meier survival plots in healthy and sick tumor-bearing mice groups, untreated or treated with saline or peptides **3** to **6**. The survival curves of peptides **3**, **4**-treated mice were significantly different, from the curves of controls, and peptides **5** and **6** treated mice. Peptides **4** and **3** significantly increased the survival of the mice with melanoma tumors from median survival time of 32–35 days (control) to 73 and 50 days (*p* ≤ 0.0001, Log-rank Mantel–Cox test), respectively (Appendix A). All mice treated with these peptides did not show any clinical signs of disease at the end of 80 days and showed no metastases in the lung at necropsy. Overall, these data indicate that MABA-peptide **4**, and to a lower efficacy MABA-peptide **3** treatment, induced a significant increase in the survival of tumor bearing mice. To confirm the anti-angiogenic effect of peptide **4**, melanoma tumor tissue sections were immuno-stained with a CD31 (platelet endothelial cell adhesion molecule 1 (PECAM-1) is a blood vessel marker) antibody. As shown in Figure 3C, fewer CD31-positive micro-vessels were observed in the tumors of mice treated with peptide **4** compared to control. These data indicate that MABA-peptide **4** significantly inhibited melanoma growth and angiogenesis in mice. Because tumor growth and angiogenesis affect each other, the question arises as to whether inhibition of angiogenesis by MABA-peptide **4**, as also found by tube formation assay in vitro (Figure 2), directly contributes to the anti-melanoma effects of peptide **4**. The answer to this question requires further experimentation. These results also indicate that, in spite of the low affinity of MABA-peptide **3** and **4** to the α1β1integrin receptor binding and cell adhesion (Table 1), using a high dose of these peptides positive anti-angiogenic and anti-tumor effects can be generated, encouraging further optimization of these lead peptides.

### 2.6. Safety of MABA-Peptide **4**

To corroborate that the anti-angiogenic effect of peptide **4** is not attributable to cytotoxicity to endothelial cells, HUVEC cell cultures were treated for 7 days with 25 mM peptide **4**, and the quantity of LDH (lactate dehydrogenase) release in the medium was measured. No significant release of LDH over the control (less than 4%) was found, indicating absence of necrotic cell death. Considering that the IC_50_ value for inhibition of cell adhesion is 0.55 mM, we trust that the therapeutic index of peptide **4** in vitro is higher than 50, and that it has high safety if endothelial cells are exposed to it.

To investigate the safety of peptide **4** in mice, male mice were intra-peritoneally injected three times a week, with a cumulative dose of 90 mg/kg for three consecutive weeks (9 injections). Acute tolerability was observed, since the mice did not show weakness, paralysis or unusual motor activity, and no inappropriate behavior was observed. Cutaneous hematomas encircling the injection or at distant locations were not observed within one day after injection. Furthermore, no sudden deaths or infections occurred during the 3 weeks of observation. Body weight increased significantly from 18 to 21 g, as in the untreated mice. After 10 h, the blood of mice injected with peptide **4** was submitted for laboratory analysis. The values for leukocytes, erythrocytes, and platelet counts were in the normal range. Alkaline phosphatase and LDH values were in the range of 100–215 units/L and 1000–2500 units/L, respectively, in blood samples of mice injected with peptide **4** and compared with control mice, ruling out any acute toxic effects on liver and other tissues. Blood urea and creatinine were in the range of 35 mg/dL and 1.2 mg/dL, respectively, and did not differ between mice injected with peptide **4** and control mice, suggesting no toxic effects to the kidney. These findings support the safety of peptide **4**.

### 2.7. Molecular Modeling

To obtain atomic level insight into the solution structures and bound structures of some of the peptides considered in this work, we subjected peptide **4** (the most active, containing two MABA units), and peptide **2** (the third most active, containing a single MABA unit, yet with a sequence highly similar to that of peptide **4**), to microsecond long MD (molecular dynamics) simulations, followed by molecular docking and MM-GBSA (molecular mechanics with generalized Born and surface area solvation) calculations. The MD simulations of both peptides were initiated from an extended conformation and were carried out for three microseconds each in an explicit solvent model. RMSD (root mean square deviation) plots (Appendix A) suggest that both simulations reached convergence at about half time during the simulations. However, surprisingly, the simulation of peptide **4** bearing the two supposedly rigidifying MABA units took longer to converge and demonstrated larger conformational changes until convergence. On the other hand, the radius of gyration (Rg) plots suggested that the solution conformation of peptide **4** is more compact than that of peptide **2** (Figure 4).

Next, the resulting trajectories were clustered, giving rise to seven clusters for peptide **4** (with 14,282, 508, 125, 66, 16, 3, and 1 members) and six clusters for peptide **2** (with 14,212, 495, 200, 77, 12, and 5 members). These numbers clearly indicate that both peptides in solution spend most of the time in a single conformation. The central structures from the largest cluster for peptides **2** and **4** are presented in Appendix A and demonstrate the compact conformation in solution of these peptides.

Next, the central structure of each cluster was docked into the crystal structure of the alpha-1 integrin I-domain, which was crystalized in complex with a collagen-mimetic peptide (PDB code 2M32) using PIPER (Schrodinger’s protein–protein docking tool). For each input structure, PIPER produces multiple clusters. The centroids of these clusters were ranked based on their PIPER scores and the best scoring pose was subjected to MM-GBSA calculations, to compare the predicted binding free energies between peptide **2** and **4** (PIPER scores can only be used to compare between different poses of the same peptide, whereas MM-GBSA energies could be used to compare the binding affinities of different peptides). The results presented in Table 2 suggest that, in agreement with the experimental findings, peptide **4** is a better binder to the integrin than peptide **2**.

Figure 5 presents the binding modes of the two lowest energy poses of peptides **2** and **4** at their predicted integrin binding sites as obtained from the MM-GBSA calculations, and Table 3 provides a summary of their interactions with binding site residues. Both peptides bind the integrin in the vicinity of the collagen-mimetic peptide binding site at two different regions, albeit with some overlap. Thus, this overlap region (around Asp177) might serve as a hotspot for the design of additional integrin KTS-ligands. Peptide **2** makes more interactions with the surface of the integrin, covering a larger region, whereas peptide **4** interacts with a smaller region of the integrin, but “penetrates” deeper into the binding site, in particular in the vicinity of Trp1. The rather small overlap between the binding sites of the two peptides might seem at first glance surprising, in light of the high sequence similarity between them. We note, however, that the presence of a single MABA group in peptide-**2** vs. the two MABA groups in peptide-**4**, led to substantial conformational differences between the two peptides, both in their unbound and bound conformations. Thus, the RMSD difference between the centroids of the two largest clusters obtained from the solution simulations of the peptide (based on the backbone of their common sequences) is 6.7 Å, whereas the RMSD difference between their bound conformations (calculated in the same way) is 7.9 Å.

A comparison between the solution conformations and the bound (i.e., bioactive) conformations of the two peptides is informative. For peptide **2**, the RMSD values between the docked conformation and the six cluster centers are 6.2 Å, 3.7 Å, 4.3 Å, 5.8 Å, 5.6 Å, and 7.2 Å. For peptide **4**, the RMSD values between the docked conformation and the seven cluster centers are 8.2 Å, 7.3 Å, 6.1 Å, 11.3 Å, 5.7 Å, 8.8 Å, and 7.7 Å. Moreover, for both peptides, the lowest energy-binding pose (based on the MM-GBSA energies) came from one of the least populated solution clusters (clusters 2 and 3 for peptide **2** and **4**, respectively). Taken together, these data suggest that the two peptides had to undergo a substantial conformational change to bind the integrin, a change accompanied by an enthalpy “penalty”. This in turn may indicate that yet more peptides that are active could be designed by further constraining their unbound conformations to the respective docked conformations. Finally, we calculated the RMSD differences between the solution conformations of peptide-**2** and peptide-**4** (represented by the centroid of their largest cluster) and a representative conformation of viperistatin taken from the MD simulation reported in reference [14]. Viperistatin is a 41-resudues peptide snake venom toxin, known to strongly and selectively inhibit α1β1 integrins. While a direct comparison between the peptides considered in this work and viperistatin is hampered by the large differences in their sizes, the results (RMSD values of 7.0 Å and 6.6 Å for peptide-**2** and peptide-**4**, respectively, based on the backbone sequence common to viperistatin and the two peptides), suggest that both peptides share some structural similarity with the more potent compounds. Interestingly, the more active peptide (peptide-**4**) is slightly more similar to viperistatin than the less potent compound (peptide-**2**), although the difference is not significant.

## 3. Discussion

Herein, we described the synthesis, structure–activity relationship, and molecular modeling of synthetic linear KTS-peptides containing meta-aminobenzoic acid (MABA) as lead antagonists of α1β1 integrin. The collagen-binding integrins (α1β1, α2β1, α10β1, and α11β1) contain a 200 amino-acid domain (I- or A-) between the second and the third calcium-binding motifs within the amino-terminal of the α-subunit that plays a key role in collagen-binding [34]. The αI-domain-containing integrin α1β1 is a high-affinity receptor for collagen type IV, and binds, albeit with less avidity, to other collagens, such as type I, XIII, and XVI [35]. Integrin α1β1 is highly expressed on micro-vascular endothelial cells and blocking its adhesive properties to collagen IV significantly reduced vascularization and tumor growth in the mouse [10,19,20,21], suggesting potential novel drug development strategies that target these receptors for the treatment of oncological diseases. Thus, except the α2β1 integrin, which is targeted by several C-type lectin-like proteins, the KTS (Lys-Thr-Ser) and RTS (Arg-Thr-Ser) viperistatin and obtustatin are 41-residue monomeric, small disintegrins that show very high selectivity to α1β1 integrin only, in contrast with RGD disintegrins, which block the integrins α3β1, α5β1, α8β1, αvβ1, αvβ3, and αlIbβ3. The selective recognition of the α1β1 integrin by KTS disintegrins is strictly dependent on the KTS motif, because mutations of individual amino acids in this motif decreased potency by 80-fold [20]. Nuclear magnetic resonance studies of the solution structure [16] and internal motions [36] of obtustatin have revealed that, in contrast with all known disintegrin structures, in which the RGD adhesion motif is located at the apex of an 11-residue hairpin loop, the active KTS motif is oriented towards one side of a 9-residue integrin-binding loop. The KTS integrin-binding loop of obtustatin exhibits lateral local motions within a range of approximately 35° and with maximal displacement of approximately 5 Å, and the integrin-binding loop and the C-terminal tail of the disintegrin display concerted motions in the 100–300 psec timescale. The shape and size of the short integrin-binding loop, along with its composition, partially restricted conformation and distinct contact with the α1β1 integrin, may underlie the structural basis of the unique selectivity and specificity of KTS-disintegrin and derived KTS-peptides for integrin α1β1. Based on these considerations, we used those disintegrins as lead compounds for the synthesis of peptides containing the KTS motif in conformational constraint by introducing one, two, and three MABA units in the vicinity of the KTS motif, hoping to preserve selectivity, to restrict conformation and provide metabolic stability. The impact of MABA location around the KTS motif, and the numbers of MABA, on the binding of GST-α1-A domain to collagen IV fragment CB3, and α1-overexpressor cells adhesion to collagen IV indicated that peptide **4** containing two MABA units was the most potent, albeit with very low avidity, and suggested recognition of a binding motif in the α1-A domain. Peptide **4** showed an outstanding selectivity towards the α1β1 integrin, about five-fold lower activity towards α2β1, and did not show any inhibitory activity toward α6β1, α5β1, α4β1, α9β1, αvb3, and αIIbb3 integrins. These findings demonstrate that integrin target selectivity can be preserved, as in the parental disintegrins, by switching from proteins to small MABA- peptidomimetics, but further optimization using cyclization and assisted by computational techniques, such as docking and MD simulations [14,21], is recommended to improve their binding avidity to the sub-micromolar range for pharmaceutical purposes.

MABA-peptides **4** and **2** containing two and one MABA units, respectively, demonstrated metabolic stability in rat and human serum. Peptide **4** containing two MABA units, and to a lower extent peptide **3** containing one MABA unit, exhibited the greatest inhibitory effect in blocking VEGF-induced angiogenesis in the Matrigel-tube formation in vitro assay, and significantly increased survival of mice with melanoma tumors. Peptide **4** was found safe in mice upon intra-peritoneal injection.

Molecular dynamics simulations followed by molecular docking and MM-GBSA calculations indicated that peptide **4** bearing the two rigidifying MABA units took longer to converge, demonstrated larger conformational changes until convergence, and spent most of the time in water in a single conformation. By the reduced conformational flexibility of this peptide, induced by the insertion of two MABA units, the radius of gyration plots suggested that the solution conformation of peptide **4** is more compact than that of peptide **2**, a property that may affect its metabolic stability and binding to the integrin. Representative solution conformations of peptides **4** and **2** were docked into the crystal structure of the α1 integrin I-domain, which was crystallized in complex with a collagen peptide using Schrodinger’s protein–protein docking tool (PIPER). The centroids of the resulting pose clusters were ranked based on their PIPER scores, and the best-scoring poses were subjected to MM-GBSA calculations to compare the predicted binding free energies. The results indicated that peptide **4** is a slightly better binder to the integrin than peptide **2** (MM-GBSA energies of −91.1 and −88.1 kcal/mol for peptide **4** and **2**, respectively) in accordance with the experimental findings. Despite their high sequence similarity, peptides **4** and **2** differ both in their unbound and bound conformations (RMSD differences of 6.7 Å and 7.9 Å between the unbound and bound conformations, respectively; see above.). Thus, the differences between their binding sites are likely to result from their different bound conformations, whereas the differences in their binding affinities are likely to result from their different binding sites, as well from differences in their unbound conformation, leading to different energetic penalties incurred by the peptides upon going from the unbound to the bound state. Finally, the small overlap in their binding region, together with their distinct binding site interactions, may provide valuable hints as to how these peptides could be modified to improve their activities. While additional pre-clinical characterization is required, MABA peptide **4** can well serve as a viable starting point for the rational design of an improved α1β1-selective peptidomimetic antagonist drug.

## 4. Materials and Methods

### 4.1. Materials

Collagen IV (from bovine placenta villi) and vitronectin were purchased from Chemicon (Temecula, CA, USA), and collagen I (from rat tail) and Matrigel from BD Biosciences (Bedfor, MA, USA). Human fibronectin and laminin were purchased from Sigma-Aldrich (St. Louis, MO, USA) and 96-well polystyrene radioimmunoassay plates were obtained from Nunc (Roskilde, Denmark Bovine serum albumin (BSA) and Hank’s Balanced Salt Solution (HBSS) sulfate was purchased from Sigma-Aldrich (St. Louis, MO, USA) [14]. The celltracker™ green 5-chloromethylfluorescein diacetate (CMFDA), was purchased from Invitrogen-Molecular Probes (Eugene, OR, USA), rabbit polyclonal antibodies against GST were purchased from Molecular Probes (Nijmegen, The Netherlands), and alkaline phosphatase-conjugated anti-rabbit antibody and p-nitrophenyl phosphate were from Sigma-Aldrich (St. Louis, MO, USA) [14]. Viperistatin was purified from the venom of *Vipera xantina palestinae* as previously described [10].

### 4.2. Cell Lines

Human umbilical vein endothelial cells (HUVECs) were obtained from Lonza Bioscience (Haifa, Israel) and B16 melanoma was purchased from the American Type Culture Collection (Manassas, VA, USA).

### 4.3. Peptide Synthesis Reagents and General Procedure for Peptides Preparation and Characterization

#### 4.3.1. General

All protected amino acids and Wang resin were purchased from GL Biochem Ltd. (Shanghai, China). *N*,*N*-Diisopropylethylamine (DIPEA), 2-(1*H*-benzotriazole-1yl)-1,1,3,3-tetramethyluronium hexafluorophosphate (HBTU), and 1-hydroxybenzo-triazole (HOBt) were purchased from BioLab Ltd. (Jerusalem, Israel), and all coupling reagents, chemicals, and solvents of a high grade were purchased from Sigma-Aldrich. The peptidomimetics were synthesized on a solid support by the standard Solid Phase Peptide Synthesis (SPPS) fluorenyl-methyl-oxy-carbonyl (Fmoc) methodology [14]. The synthesis was carried out manually on Wang resin using Fmoc-protected amino acids. Meta-aminobenzoic acid (MABA) was also Fmoc protected. Coupling was performed for 1 h with four equivalents of TBTU and one equivalent of amino acid in the presence of eight equivalents of DIEA. Fmoc groups were removed with 20% piperidine in DMF. Washings after Fmoc removal and couplings were performed three times with DMF. After the final assembly, the peptidyl-resin was washed three times with DMF, three times with DCM, and three times with MeOH. The resin was then dried 3 h in oven at 60 °C. Cleavage from the resin and full deprotection of peptides was performed using a mixture of trifluoroacetic acid (TFA)/phenol/water/tri-isopropyl-silane [88:5:5:2 (*v*/*v*/*v*/*v*)] for 3 h at room temperature [37]. The resin was removed by filtration and the residual peptides were precipitated by addition of cold diethyl ether to the filtrate. The precipitate was separated by centrifugation at 3500 rpm for 10 min, dried under vacuum, solubilized in water, and lyophilized. The synthesized peptides were purified by preparative reverse-phase (RP)-HPLC using either Kromasil and NanoChrom, ChromCore 120 C18 HPLC columns (5 µm particle size, pore size 100 Å, 250 mm × 4.6 mm, Sigma-Aldrich), with an elution gradient of 0–100% acetonitrile, with 0.1% trifluoroacetic in 100% water, at a flow rate of 1.0 mL/min, using a Dionex UltiMate 3000 analytical HPLC (Thermo Scientific, Waltham, MA, USA), with detection at a wavelength of 220 nm. The solid phase peptide synthesis (SPPS) was carried out according to the Fmoc method applying Wang resin (1.5 mmol/g) as a solid support.

#### 4.3.2. Loading

0.1 mmol of the Wang resin was swelled for 1 h in NMP and drained by filtration. Then, 0.5 mmol (5 eq) of FmocAa and 0.5 mmol (5 eq) of HBTU were dissolved in 2 mL of NMP + 1 mmol (10 eq) of DIEA (10 eq = 1.74 mL) and stirred for 3 min. Subsequently the solution was added to the resin. The resin was then vortex with for 3 h at RT, filtrated and washed with NMPx4 and DCMx4.

#### 4.3.3. Fmoc Deprotection

Removal of the Fmoc protecting group was implemented by incubating the pre-swollen resin with 2 mL of 20% piperidine in NMP for 15 min. The solution was then filtered, and the resin washed with NMP (2 × 2 mL). The piperidine step was repeated once again, and the resin was washed with NMP (4 × 2 mL) and DCM (4 × 2 mL).

#### 4.3.4. Elongation with the FmocAa

The reaction progress was checked by the Kaiser test. Finally, after the last deprotection step, the resin was washed with NMP (4 × 2 mL), DCM (4 × 2 mL), MeOH (1 × 2 mL), DCM (4 × 2 mL), then dried under vacuum.

#### 4.3.5. Cleavage from Resin

TFA/Et_3_SiH (90/10), 1.5 mL, was added to the resin and vortexed for 3.5 h. Then the solvent was removed by filtration and the resin was washed again with TFA/H_2_O (95/5, 1 mL). The TFA solutions were collected and the peptide was precipitated from the solution by addition of cold Et_2_O. The solid precipitation was washed with cold Et_2_O and dried. The synthetic peptide was dissolved in distilled water, lyophilized, and applied to HPLC for purification. The collected fractions were further analyzed by LC-MS.

#### 4.3.6. Mass Spectroscopy

The single-quad liquid chromatograph ultra-fast mass spectrometer LC-MS Model 2020 (Shimadzu, Kyoto, Japan), composed of a binary pump (20 AD), vacuum degasser, thermostatic auto sampler (SIL 20ACHT), thermostatic column compartment (CTO 20A), photodiode detector (SPD M 20A) and mass analyzer (MS 2020), with both electrospray ionization (ESI), and atmospheric-pressure chemical ionization (APCI) systems. Optimized mass spectra were acquired with an interface voltage of 4.5 kV, a detector voltage of 1.2 kV, a heat block temperature of 300 °C and a desolvation gas temperature of 200 °C. Nitrogen was used as the nebulizer gas at a flow rate of 1.5 L/min and dry gas flow of 10 L/min. LC-MS data acquisition and processing was performed by a single method that quickly review the ESI and DUIS results in the data browser window (https://www.shimadzu.com/an/products/liquid-chromatograph-mass-spectrometry/single-quadrupole-lc-ms/lcms-2020/spec.html, last accessed on 7 April 2024).

### 4.4. ELISA to Test GST-α1A Binding to Collagen IV Fragment CB3

The enzyme-linked immunosorbent assay (ELISA) was performed as previously described [38], with the following modifications: CB3 (collagen IV fragment) was immobilized overnight at 4 °C on a microtiter plate at 10 mg/mL in Tris-buffered saline (TBS)/MgCl_2_ (50 mM Tris-HCl, 150 mM NaCl, and 2 mM MgCl_2_, pH 7.4) and 0.1 M acetic acid, respectively. After blocking the non-specific binding of the plate with 1% BSA in TBS/MgCl_2_, the GST-tagged α1A-domain was allowed to bind to collagen IV-fragment CB3 in the presence or absence of different concentrations of peptides, for 2 h at room temperature. The bound GST-α1A domain was fixed for 10 min with 2.5% glutaraldehyde in HEPES buffer (50 mM HEPES, 150 mM NaCl, and 2 mM MgCl_2_, and 1 mM MnCl_2_, pH 7.4). The bound GST-α1A was quantified with rabbit polyclonal antibodies against GST, followed by alkaline phosphatase–conjugated anti-rabbit antibody, which served as the primary and secondary antibodies, respectively, each diluted in 1% BSA in TBS/MgCl_2_. The conversion of p-nitrophenyl phosphate was measured at 405 nm using an enzyme-linked immunosorbent assay reader (BioTek, Bad Friedrichshall, Germany). Nonspecific binding was assessed by binding of GST-α1A to BSA in the presence of 10 mM EDTA [14,38]. Inhibitory concentration 50% (IC_50_; mean ± SEM) was calculated using a non-linear regression algorithm from three repetitive experiments generating titration inhibition curves performed in triplicates *(n* = 9 experimental points at each concentration) [39].

### 4.5. Cell Adhesion Assay

The assay was accomplished as reported previously [30]. The day before the experiment, the plate was coated with 10 µg/mL collagen IV or 1 µg/mL collagen I in 0.02 M acetic acid, and incubated overnight at 4 °C. Thereafter, nonspecific binding was blocked by incubating the wells with 1% (*w*/*v*) BSA in HBSS containing 5 mM magnesium chloride (MgCl_2_), at room temperature for 1 h before use. The cells were labeled by incubation with 12.5 µM CMFDA in HBSS without 1% BSA at 37 °C for 30 min. The labeled cells were then centrifuged at 1000 rpm, and washed twice with HBSS containing 1% BSA, to remove excess CMFDA. Labeled cells (1 × 10^5^ cells/well) were added to each well, in the presence or absence of peptides, and incubated at 37 °C for 60 min. In the presence of peptide, the cells were added to the well after prior incubation with peptide for 30 min at 37 °C. Unbound cells were removed by washing the wells three times with 1% (*w*/*v*) BSA in HBSS, and bound cells were lysed by the addition of 0.5% Triton X-100 (diluted in double-distilled water). The fluorescence in each well was measured with a SPECTRAFluor Plus plate reader (Tecan, San Jose, CA, USA) at an excitation wavelength of 485 nm and an emission wavelength of 530 nm. To determine the total number of adhered cells using the fluorescence values, a standard curve was generated using serial dilutions with known numbers of CMFDA-labeled cells. Inhibitory concentration 50% (IC_50_; mean ± SEM) was calculated from three experiments performed in triplicate (*n* = 9 experimental points at each concentration), compared to cells grown on a gelatin-coated wells (gelatin was used as control for collagen) [30].

### 4.6. Stability of the Peptides in Rat Serum

1.5 mL of fresh plasma obtained from male Wistar rats (Harlan, Israel) was incubated at 37 °C with a stock solution, to make a final peptide concentration of 10 µg/mL. Metoprolol (1 µg/mL) was used as an internal standard. Samples in triplicate were taken at time 0 and after 5, 10, 20, 30 and 60 min, and 50 µL of the reaction solution was removed and added to 100 µL of ice-cold acetonitrile. The samples were vortex-mixed for 1 min and then centrifuged at 14,600× *g* for 10 min to precipitate the serum proteins. The supernatant was then transferred to fresh glass tubes and evaporated to dryness (Vacuum Evaporation System, Labconco, Kansas City, MO, USA). Then, the glass tubes content was reconstituted in 80 μL of a solution of 50% water and 50% acetonitrile supplemented with 0.1% formic acid and centrifuged for a second time (14,600× *g*, 10 min). The resulting solution was injected (10 μL) into the HPLC-MS device (HPLC-MS Waters 2695 Separation Module, equipped with a Micromass ZQ detector). The column used was Kinetex^®^ (Phenomenex, Torrance, CA, USA) 2.6 μm EVO C18 100 Å, LC column 100 mm × 2.1 mm. A linear gradient from 80% buffer A (0.1% formic acid in water), to 10–90% buffer A and buffer B (0.1% formic acid in acetonitrile), was applied for 15 min. The flow rate used was 0.2 mL/min at 25 °C as previously described [40].

### 4.7. Tube Formation-Angiogenesis Assay

Tube formation assay was performed as previously described [41]. Briefly, human umbilical vein endothelial cells (HUVEC) were serum-starved in medium (PeproGrow endothelial cell media, PeproThec Co., Rocky Hill, NJ, USA) for 3 h before applying to Matrigel^®^ polymerized in Ibidi angiogenesis μ-slides and 96-well plates at 37 °C for 1 h. 2 × 10^4^ cells per well were treated with 100 µg/mL of the tested peptide or left without treatment (control). The cells were cultured at 37 °C, with 5% CO_2_ for 6 h, in the presence of a complete, chemically defined formulation medium for the in vitro cultivation of endothelial cells from large blood vessels and angiogenesis VEGF inducer (PeproGrow MacroV, PeproThec Co., Rocky Hill, NJ, USA), and then analyzed by light microscopy (IX-73, OLYMPUS, Tokyo, Japan). The quantitation of tube length, nodes, branches, and mesh was performed with the angiogenesis analyzer for Image J software (Version 1.53t, NIH Image company, Bethesda, MD, USA) [33].

### 4.8. In Vivo Characterization of Anti-Tumor Activity

C57BL/6 young male mice (aged 4–6 weeks, weighing 25 g) were obtained from Envigo Ltd. (Rehovot, Israel), and used after acclimation for four days to laboratory conditions. During the study, the mice were housed within a specific pathogen-free animal house facility, and kept in groups with a maximum of seven mice per cage (Polypropylene, Euro standard type IV, floor area of the cage: 425 × 266 × 185 mm (800 cm^2^), fitted with solid bottoms, and a static cage filter top (Tecniplast Co., Buguggiate, Italy), filled with 7090 Teklad sani-chips animal bedding (Envigo Ltd., Rehovot, Israel), and having two plastic tubes in each cage as enrichment material. Three days prior to implantation of tumor cells, the mice were provided ad libitum with a commercial Teklad 2018S global 18% protein rodent diet (Envigo Ltd., Rehovot, Israel), had free access to drinking water, which was monitored periodically and supplied to each cage via polyethylene bottles with stainless steel sipper tubes. The mice were housed in controlled environmental conditions, and the temperature was maintained at 17–23 °C, with a relative humidity of 30–70% on a 12-h light/dark cycle and 15–30 air changes/h. The mice were assigned to experimental groups using an online random number generator (https://www.graphpad.com/quickcalcs/randomize1/ (accessed on 10 June 2021)), assigning seven mice subjects to a group. The experiments included seven groups (a. Healthy-Saline, Control; b. Sick-untreated, Control; c. Sick-Saline, Control; d–h. MABA-peptide **3** to **6**-treated sick mice) and was repeated twice during a 12-month period. Cages were likewise allocated to treatment groups with randomly generated numbers. Body weights were measured throughout the study. Two female experimenters were blinded to the allocation of the mice and conduct of the experiments, and another female investigator was blinded for the outcome assessments and data analysis. All procedures were carried out in compliance with the ARRIVE guidelines and adhered to the guidelines of the joint ethics committee (IACUC) of the Hebrew University (an AAALAC International accredited institute), and Hadassah Medical Center in Jerusalem, Israel, which approved the study protocol for animal welfare (MD-09-11910-5).

The B16 murine melanoma cell line was maintained in RPMI 1640 (Gibco BRL, Grand Island, NY, USA) supplemented with 10% fetal bovine serum (FBS; Sigma, St Louis, MO, USA), 20 mM HEPES (Gibco BRL), penicillin and streptomycin 100 IU/mL and 100 µg/mL, respectively. Cells in the log-phase growth (24–48 h after plating) were collected by brief treatment with trypsin and washed with phosphate-buffered saline (PBS). The cells were first incubated with the peptides in PBS for 2 h at 37 °C or left untreated (control), and thereafter 7 × 10^5^ cells were injected subcutaneously in 0.15 mL PBS in the right flank of the mice. Seven to eight mice were used for each group examined. The peptides were administered three times a week, for three weeks after cell inoculation, by intraperitoneal injection of a 10 mg/kg dose (cumulative dose of 90 mg/kg/mouse) in 0.2 mL PBS. Mice survival was followed-up every day for 80 days after cells’ inoculation and the death was recorded. Toxicity was evaluated by mice examination and by body weight assessment. In the case that a mouse showed clinical signs of discomfort or pain, it was euthanized and necropsy was performed. Mice survival was analyzed by the Kaplan–Meier method. The median survival of groups of B16 melanoma mice were compared to control group of sick tumor-implanted but non-peptide-treated mice with GraphPad Prism 5, performing both the log-rank (Mantel-Cox) test and the Gehan–Breslow–Wilcoxon test (Appendix A). Experiments were performed under double blind testing. Palpable tumors under the skin were measured in two perpendicular diameters using a digital caliper. Tumor volume (V), which correlates well with tumor weight, was calculated as V = 0.5 LW^2^ (L is the largest length diameter and W is the smallest width diameter). Sixteen days after implantation of the cells, mice were killed by pentobarbital anesthesia (120 mg/kg) and tumors were excised and weighed. Some mice were killed 3 days after implantation of the cells to determine tumor size, and at this time, tumors were about 2 mm diameter and therefore were not yet dependent on the angiogenic process. Sections of tumors were formalin fixed for further study. Immunohistochemical analyses of blood vessel formation were performed with goat anti-mouse CD31 antibody (Santa Cruz Biotechnology, Santa Cruz, CA, USA) using the labeled streptavidin-biotin method. Briefly, sections were deparaffinized in xylol and rehydrated in a graded alcohol series. Antigen retrieval was carried out by autoclaving sections in retrieval buffer (10 mM EDTA citrate buffer, pH 6.0) for 3 min. Endogenous peroxidase activity was blocked by incubation in 3% hydrogen peroxide at room temperature in the dark for 20 min. Non-specific binding of reagents was reduced by incubation of sections for 30 min in 5% normal rabbit serum. Sections were then incubated with goat anti-mouse CD31 (dilution 1/200) antibody overnight at 4 °C, followed by incubation with biotinylated rabbit anti-goat IgG, and then streptavidin-biotin-horseradish peroxidase complex at 37 °C for 1 h. A negative control was included with each run by replacing the primary antibody for non-immune rabbit serum. The nuclei were counter-stained with hematoxylin. Representative images were taken under a light microscope (×400) in randomly selected fields, and the number of blood vessels stained brown were counted. Statistical analysis of the differences in tumor volume and blood vessel density were performed using one-way analysis of variance (ANOVA) and a value of *p* < 0.05 was considered statistically significant.

### 4.9. Cell Death

Cell death was assessed by measuring the release of lactate dehydrogenase (LDH) into the medium using the Lactate Dehydrogenase necrotic assay [42]. LDH activity was measured spectrophotometrically at 340 nm by following the rate of conversion of oxidized nicotinamide adenine dinucleotide (NAD) to the reduced form of NADH. LDH release was expressed as the optical density (OD) units.

### 4.10. Toxicity to Mice

Experiments with animals and animal care were approved by the Hebrew University Committee of Ethics and were performed in strict accordance with the Guide for the Care and Use of Laboratory Animals published by the US National Institutes of Health. Adult male C57BL/6 mice (*n* = 10) weighing 18 g, and aged 4 weeks, were housed in temperature-controlled rooms (22–25 °C), with access to water and food ad libitum. Five mice were intraperitoneal injected twice a week, with a dose of 50 mg/kg of peptide **4** for four consecutive weeks (8 injections) and the control group of five mice were injected with saline. The animals were daily examined for autonomic symptoms, paralysis, motor activity, and regular behavior. Blood samples were taken from control and peptide-injected mice after 10 h from injection and were analyzed for hematocrit and biochemical analysis at Koret School of Veterinary Medicine, Faculty of Agricultural, Food and Environment, The Hebrew University of Jerusalem at Rehovot, Israel.

### 4.11. Molecular Dynamic Simulations

The starting structures for the simulations of compounds **2** and **4** were built in Maestro 3D Builder as extended peptides and the two termini were kept charged. The structures were minimized using the protein preparation protocol as implemented in Maestro (Schrödinger Release 2022-2) [43] and proper protonation states of residues were determined. Quantum mechanical calculations were performed for the minimized conformation of the compounds to calculate atomic charges to be used in molecular dynamics (MD) simulations. Geometry optimization was performed for each compound, and the electrostatic potentials were calculated at the HF/6-31G* level using the Gaussian 09 software. The partial atom charges were determined using the RESP method [44] with the AnteChamber software [45,46]. MD simulations were carried out using the AMBER99SB-ILDN force field [47] as implemented in the Gromacs program (v. 2021.1) using GPU accelerators [48,49,50]. The compounds were first minimized in a cubic box of TIP3P water molecules with an extra extension along each axis of the peptide of 10 Å under periodic boundary conditions. Ions were added to the solution to make the system electrically neutral. After the minimization, compounds were equilibrated first under NVT conditions (at T = 300 °K for 10 ns) employing the V-Rescale thermostat [51] and subsequently under NPT conditions (at T = 300 °K and *p* = 1 bar for 10 ns) employing the V-Rescale thermostat [51] and the Parrinello-Rahman barostat [52]. The simulations were run with a time step of 2 fsec using the leap-frog algorithm. Long-range electrostatic interactions were computed using Particle Mesh Ewald Summation [53,54]. All simulations employed a 10 Å cutoff for van der Waals and Coulomb interactions using the Verlet cutoff scheme [55]. The LINCS algorithm was used to constrain bond lengths [56]. Production runs were performed under NPT conditions as described above and were run for 3 μsec each [57]. Root Mean square deviation (RMSD) calculations indicated that both simulations are converged (Appendix A).

To explore the conformational space of the compounds, we performed clustering of 15,000 frames retrieved from the MD production run using the Gromos method [58] as implemented in the Gromacs software version 2021.1. The algorithm counts the number of neighbors of each structure using a cut-off (0.5 nm), forms a cluster from the structure with the largest number of neighbors with all its neighbors and eliminates them from the pool of structures. The process was iterated until all structures have been considered. This procedure resulted in a total of 6 and 7 clusters for compound **2** and **4**, respectively. The most populated cluster for all compounds was the first one, including 14,212 conformers in case of compound **2**, and 14,282 conformers in case of compound **4**. The central structure (i.e., the structure with the smallest average RMSD from all other structures of the cluster) of the biggest cluster of compounds **2** and **4** are presented in Appendix A.

### 4.12. Molecular Docking

To gain atomic-level insight into potential binding modes of peptides **2** and **4** to the integrin I-domain and to obtain approximated binding free energies, the central structure of each cluster, was docked into the crystal structure of the alpha-1 integrin I-domain which had crystalized in complex with a collagen-mimetic peptide (PDB code 2M32 [59]). Prior to docking, the protein structure was prepared with the Protein Preparation Wizard [43,60] to assign correct protonation states for all residues at physiological pH and to add missing hydrogens. Docking was performed by means of the PIPER algorithm, a protein-protein docking tool [60,61] which is implemented in the Schrödinger software. The algorithm rotates the ligand into a large number of different orientations with respect to the receptor, and each of these orientations is scored by the PIPER’s scoring function. Finally, PIPER clusters the resulting poses based on their RMSD values and outputs the centroids of the clusters with the respective PIPER scores.

### 4.13. MM-GBSA Calculations

MM-GBSA calculations were performed using Schrodinger’s Prime module with default parameters. Protein flexibility was considered using Hierarchical Sampling method in radius 3A from the peptide.

### 4.14. Statistical Analysis

Statistical analyses were performed using the software GraphPad Prism version 8.0.2 (GraphPad Software Inc., San Diego, CA, USA). Statistical significance was determined by the one-way analysis of variance (ANOVA) or Student *t*-test. Median survival of mice was analyzed by Log-rank Mantel-Cox test. Normal distribution was confirmed using Shapiro-Wilk test (α = 0.05). The experiments were performed three times, and results are expressed as mean ± SEM. The differences were considered significant at *p* ≤ 0.05.

## 5. Conclusions

In summary, we have reported the synthesis and characterization of meta-aminobenzoic acid (MABA)-KTS-peptides as antagonists of α1β1 integrin I-domain. The findings indicate that the number of MABA units and their location in the KTS peptide caused different binding potencies, MABA-KTS-peptides are metabolically stable and safe, and that MABA-peptide **4**, bearing two MABA units had the most anti-angiogenic and anti-tumor activity. Molecular modeling indicated that MABA-peptide **4** has a compact conformation in water and penetrates the integrin’ binding site. These results serve as a foundation for future optimization and drug development of α1β1 integrin cyclic peptide antagonists that may be used as lead compounds for developing potential therapeutic agents for angiogenesis-related diseases, including cancer.

## Figures and Tables

**Figure 1 pharmaceuticals-17-00549-f001:** Metabolic stability of the MABA-peptides in rat plasma (*n* = 3; * *p* ≤ 0.05 compared to time 0 or 10 min). The amount is presented as a percentage of the sample at time 0.

**Figure 2 pharmaceuticals-17-00549-f002:** Quantification of tube formation after treatment of HUVEC cells with MABA-peptides in the Matrigel assay. Phase contrast images taken after 6 h of the treatment at 40× magnification. The left column presents typical phase contrast images, the right column indicates the angiogenesis analyzer of quantification of tube formation as the number of junctions (top, left), nodes (top, right), meshes (bottom, left) or total tube length (bottom, right) per field of view. Error bars indicate SEM calculated from three different experiments performed in triplicate (* *p* ≤ 0.05; ** *p* ≤ 0.01; *** *p* ≤ 0.001).

**Figure 3 pharmaceuticals-17-00549-f003:** Anti-melanoma effects of the MABA-peptides in C57BL/6 mice bearing B16 melanomas. The cells (7 × 10^5^) were subcutaneously inoculated at the right flank of individual mice. Immediately after cell injection, mice were randomly divided into different groups (*n* = 8) and intra-peritoneally administered with vehicle control (PBS solution) or 10 mg/kg of MABA-peptide **3** to **6** three times a week. Tumor volume was measured at the indicated days. At day 16, mice were sacrificed and tumors were dissected; (**A**) Tumor volume of MABA-peptide **4** treated mice (light blue) compared to untreated sick group (blue); * *p* < 0.01 and ** *p* < 0.001 versus the respective sick group; (**B**) Kaplan–Meier survival plots of saline- (dashed line) or MABA-peptide-treated (solid lines) tumor-bearing mice; black-dotted line represents 50% survival; (**C**) inhibition of intra-tumoral angiogenesis. Representative images (left panel; ×400) of vascular growth, detected by immune staining of CD31 in melanoma sections from mice with different treatments. Number of brown stained, CD31-exposing vessels (encircled in yellow) of each group was calculated using NIH Image J software (Version 1.53t, NIH Image company) (right panel). Data are presented as mean ± SEM of 7 mice. ** *p* < 0.01 versus control group.

**Figure 4 pharmaceuticals-17-00549-f004:** Radius of gyration (Rg) plots for peptides 2 and 4 as a function of simulation time.

**Figure 5 pharmaceuticals-17-00549-f005:** The binding modes of the lowest energy poses (based on MM-GBSA energies) of peptides **2** and **4** in their predicted integrin binding sites, in comparison with the binding mode of the crystallized collagen-mimetic peptide. Peptides **2** and **4** are colored blue and orange, respectively. The α1 integrin I-domain protein is depicted in ribbon form, highlighted in green, while three crystallized collagen-mimetic peptides (PDB code 2M32) are distinguished by their coloring in yellow, cyan, and violet.

**Table 1 pharmaceuticals-17-00549-t001:** Inhibitory effect of the synthetic KTS(R)-peptides containing meta-aminobenzoic acid on binding of GST-α1 A domain to collagen IV fragment CB3 and on α1-over-expressor cells.

PeptideNumber	Sequence	α1β1 Binding In Vitro AssayIC_50_ (µM)	α1β1 Adhesion In Cell AssayIC_50_ (µM)
1	H_2_N-W-*KTS*-R-TSHY-**MABA**-TGKSDG-COOH	828 ± 8	≥3000
2	H_2_N-W-*KTS-***MABA**-R-TSHY-GKSDG-COOH	600 ± 10	≥3000
3	H_2_N-*KTS*-W-*KTS*-**MABA**-R-TSHY-GKSDG-COOH	442 ± 12	≥900 ± 75
4	H_2_N-W-*KTS-***MABA**-R-**MABA**-TSHY-GKSDG-COOH	324 ± 8	550 ± 45
5	H_2_N-W-*KTS*-**MABA**-RTSHY-**MABA**-TGKSDG-COOH	1000 ± 9	≥3000
6	H_2_N-W-*KTR*-**MABA**-R-**MABA**-TSHY-TGKSDG-COOH	2142 ± 12	2006 ± 96
7	H_2_N-W-*KTR*-**MABA**-R-**MABA**-TSHY-**MABA**-TGKSDG-COOH	2481 ± 21	≥5000
8	H_2_N-^20^W-*KTS*-R-TSHY-TGKSDG^36^-COOH	970 ± 7	≥3000

**Table 2 pharmaceuticals-17-00549-t002:** MM-GBSA binding free energies (in kcal/mol) for the centroid poses of peptides **2** and **4** from the corresponding clusters.

	Peptide 2	Peptide 4
Cluster No.	MM-GBSA Energy	MM-GBSA Energy
1	−35.8	−55.7
2	−88.1	−47.2
3	−39.7	−91.1
4	−70.0	−80.3
5	−68.9	−73.8
6	−41.5	−77.3
7		−52.8

**Table 3 pharmaceuticals-17-00549-t003:** Interactions formed between the lowest energy poses (based on MM-GBSA calculations) of peptides **2** and **4** at their predicted binding sites on the integrin structure *.

Peptide 4	Interactions between Peptide and Protein	Peptide 2	Interactions between Peptide and Protein
Trp1	HB with BB of Tyr 17	Trp1	
Lys2		Lys2	BB HB with Val175 and HB with Asp28
Thr3	HB with Ser176	Thr3	HB with BB of Leu143
Ser4		Ser4	HB BB of Ala174
MABA 5		MABA 5	
Arg6	** 2 HB with Asp177 **	Arg6	HB with Thr153
MABA 7		Thr7	HB with Asn174
Thr8		Ser8	HB with Thr168
Ser9	HB with Ser21	His9	
His10		Tyr10	HB with Thr183
Tyr11	HB with BB of Tyr146	Gly11	
Gly12	BB HB with Tyr146	Lys12	** HB with Asp177 **
Lys13		Ser13	
Ser14		Asp14	
Asp15		Gly15	BB HB with Glu154
Gly16			

* HB and BB are hydrogen bond and backbone, respectively. Interactions common to both peptides are highlighted in bold.

## Data Availability

The original contributions presented in the study are included in the article/Appendix A, further inquiries can be directed to the corresponding author.

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
