# Peer review of "Structure–Activity Relationship of Synthetic Linear KTS-Peptides Containing Meta-Aminobenzoic Acid as Antagonists of α1β1 Integrin with Anti-Angiogenic and Melanoma Anti-Tumor Activities"

_pharmaceuticals, 2024, doi:10.3390/ph17050549_

Round 1

Reviewer 1 Report

Comments and Suggestions for Authors

This manuscript reports the structure-activity relationship of synthetic linear KTS-peptides containing meta-aminobenzoic acid as antagonists of α1β1 in-3 tegrin with antiangiogenic and melanoma’ antitumor activities. Molecular dynamics simulations were performed that determine that the solution conformation of MABA peptide 4 is more compact, underwent larger conformational changes until convergence, and spent most of the time in a single cluster. Binding affinity of peptides has been characterized by enzyme linked immunosorbent assay. Overall, the study is well presented. Few issues need attention.

1. The similarity index report is my bigger concern. It shows 44% similarity content which must be revised to bring it to an acceptable limit.

2. Resolution of figures should be improved.

3. Delete word FIGURE in every figure as an appropriate legend is already given.

Comments on the Quality of English Language

Minor editing

Reviewer 2 Report

Comments and Suggestions for Authors

This paper investigates the design, synthesis, and biological evaluation of linear peptides containing the KTS pharmacophore motif and 3-aminobenzoic acid (MABA) for targeting integrin receptors. Peptide 4 exhibited promising antiangiogenic and antitumor activity. The manuscript is well-structured and presents comprehensive experimental data. However, clarity could be improved in certain sections, and further analysis is warranted.

  1. Can the authors provide more insight into the rationale behind selecting specific positions for introducing MABA units in the peptide sequences?
  2. How do the identified IC50 values for peptide 4 compare with those of existing integrin antagonists in preclinical studies?
  3. Could the authors elaborate on the mechanisms underlying the observed selectivity of peptide 4 towards α1β1 integrin over other integrin subtypes?
  4. Considering the potential clinical application, what are the challenges associated with scaling up the synthesis of MABA-peptide 4 for large-scale production?
  5. In the discussion of molecular modeling results, can the authors provide a clearer interpretation of the implications of the observed conformational differences between peptide 2 and peptide 4?

Reviewer 3 Report

Comments and Suggestions for Authors

This paper investigates the design, synthesis, and characterization of MABA-peptides targeting integrin receptors and explores the peptides' binding affinity, metabolic stability, anti-angiogenic, and anti-tumor effects. Results suggest peptide 4 as a promising lead compound. Overall, this manuscript presents an intriguing finding but would benefit from further clarity and detail in experimental methodologies and data analysis. However, the clarity in experimental methodologies and statistical analyses is needed:

  1. Can you provide more details on the statistical analysis used to determine IC50 values and significance in binding assays and cell adhesion experiments?
  2. How were the structures of the peptides confirmed after synthesis? Were any spectroscopic techniques employed?
  3. Could you clarify the mechanisms underlying the observed anti-angiogenic and anti-tumor effects of peptide 4?
  4. In the discussion of the molecular modeling results, can you elaborate on the potential implications of the conformational differences between peptide 2 and peptide 4 on their binding affinity and selectivity?
